# Cross-Scale Study on Lime Modified Phosphogypsum Cemented Backfill by Fractal Theory

Fengwen Zhao [1], Jianhua Hu [1,*], Yinan Yang [1], Hongxing Xiao [2] and Fengcheng Ma [2]

1   School of Resources and Safety Engineering, Central South University, Changsha 410083, China; dr_zfw94@163.com (F.Z.); 215511069@csu.edu.cn (Y.Y.)
2   Hubei Sanning Mining Co., Ltd., Yichang 443100, China; hbsnky@163.com (H.X.); 15642244149@163.com (F.M.)
*   Correspondence: hujh21@csu.edu.cn

**Abstract:** Pore structure is a critical factor affecting the strength characteristics of mine backfill materials. In this paper, based on the mechanical tests on lime modified phosphogypsum cemented backfill (LMPGCB), the microstructure of the LMPGCB sample was characterized by nuclear magnetic resonance (NMR) and scanning electron microscopy (SEM) imaging. Based on the fractal theory (FT), the values expressing the fractal characteristics of the NMR T2 spectrum and SEM images were determined. The mathematical model between the NMR-FT fractal dimension and the SEM-FT box dimension was developed to infer the overall pore characteristics from various pore characteristics. The functional relationships between the strength and pore content and dimension were established, and the effects of pore structure on the strength were discussed. The results indicate that: (1) in NMR-FT fractal dimension, the fractal effect of various types of pores is better. The fractal dimension of the small pore is between 0.86–1.38, in which the overall trend first decreases and then increases. (2) In the SEM-FT box dimension, the fractal effect of the overall pore is better. The overall trend of box dimension decreased first and then increased at each kind of magnification. Furthermore, with the increase of magnification, the box dimension decreases. (3) There is a linear direct proportional relationship between the SEM-FT box dimension and the fractal dimensions of NMR-FT of small pores. The relationship between SEM-FT box dimension and NMR-FT fractal dimensions of various types of pores conforms to the plane relationship. There is a linearly increasing relationship between dimension and pore content and a decreasing relationship between dimension and strength.

**Keywords:** cemented backfill; fractal theory; nuclear magnetic resonance; scanning electron microscopy; relational model; macro-meso characteristics

## 1. Introduction

As an important part of the mine filling system, the mechanical properties of filling material are of great significance to the mine. Among many factors affecting its mechanical characteristics, the pore structure is one. To deeply study the pore characteristics, advanced detection technology can be used to characterize it, so as to explore its influence law on the mechanical characteristic of backfill material. At present, the detection techniques of microstructure include mercury intrusion porosimetry (MIP), nuclear magnetic resonance (NMR), and scanning electron microscopy (SEM) [1–4]. MIP is used to measure the pore characteristics of the material by injecting mercury after the appearance of the sample is destroyed. NMR can determine the pore characteristics of materials in a non-destructive way. SEM can measure the structural characteristics of the material at high magnification. Zhao et al. [5] obtained pore characteristics such as pore size distribution, porosity, and pore content of filling materials by using NMR. Hu et al. [6] observed the material structure characteristics and pore structure characteristics inside the backfill material by SEM.

The quantitative characterization of pore structure is an important means to study the strength characteristics of backfill materials. The fractal theory is the theoretical basis to study the laws of objective things by mathematical methods. The complexity of things is represented by dimension. The commonly used fractal dimensions are the Hausdorff dimension and Box dimension [7]. Because the structural characteristics of pores are very complex, they cannot be described by traditional geometry, but they have certain self-similarity, and fractal theory is usually used to analyze complex geometry with self-similarity structure [8–11]. Therefore, using fractal theory to analyze pores is a good choice. By combining the fractal theory with NMR or SEM, the pore dimension characteristics can be obtained to quantitatively analyze the characteristic pore parameters. The fractal dimension characteristics of various types of pores can be well-obtained [10–12]. Hu et al. [13] found that all kinds of pores of backfill materials had a good fractal effect when they studied backfill materials using NMR. Deng et al. also used the combination of fractal theory and NMR to study the influence pore on the mechanical characteristics of backfill [9]. When the fractal theory was combined with the electron microscope, the dimension characteristics of the overall pores of the samples could be well-analyzed [14,15]. Wang et al. [16] showed that the box dimensions could be used to describe the pore characteristics of diatomite. As there is little difference between the peaks of the NMR spectrum of dense materials, the fractal dimensions of the whole can be obtained [17].

However, for porous materials, the peaks of the NMR spectrum differ significantly, resulting in a poor fractal effect of the whole. Thus, it becomes difficult to achieve a comprehensive analysis from the whole to the part. Phosphogypsum and lime participate in hydration reaction, which can be the formation of hydration products. Thus, backfill materials such as phosphogypsum and lime (base) were used as the primary research focus, and tailings, tailing sludge, fly ash, and cement were added to prepare the backfill, in this study. A uniaxial compressive strength (UCS) test was carried out, and the resulting microscopic characteristics of the backfill were obtained by NMR and SEM. The fractal characteristics and mathematical relationships for different types of pores in backfill materials are obtained. The functional relationship model of macro strength is discussed and established to study the cross-scale characteristics of the backfill materials comprehensively.

## 2. Materials and Methods

### 2.1. Materials

The tailings, tailing sludge, phosphogypsum, and fly ash were all obtained from Sanning Mining. Ordinary Portland cement (OPC) was obtained from the Changsha Xinxing Cement Factory, Changsha, China. Commercially available lime containing 85% CaO was used. Tap water was used in the experiment. Material characterization was carried out by sieve analysis (JTG E42-2005), laser particle size analyzer (Mastersizer 2000, Malvern Instruments Ltd., Malvern, UK), X-ray fluorescence spectroscopy (XRF, PANalytical B.V., Panaco, The Netherlands), and X-ray diffraction (XRD, PANalytical B.V., Panaco, The Netherlands). The particle size distribution, other physical properties, and mineral composition are given in Figure 1 and Tables 1–3.

### 2.2. Experiment Methods

#### 2.2.1. Experimental Design

Tailings and tailing sludge were used as filling aggregate, while OPC and fly ash were used as binders, and lime was used as the phosphogypsum activator. The manufacturing and curing process of the backfill sample is as follows. Standard samples in 50 mm × 100 mm cylindrical molds were cast from a slurry of different proportions. They were labeled as B and A1 through A7 in the sequence according to the proportion. Nine specimens in each group were prepared. Thus, there were a total of 72 specimens. The mixture proportions are given in Table 4, in which the ratios of tailing sludge to the tailings and fly ash to tailing sludge are 1:3 and 1:2, respectively. The samples were cured in the curing box (20 °C and 99% humidity). All tests were carried out at the 28-day age.

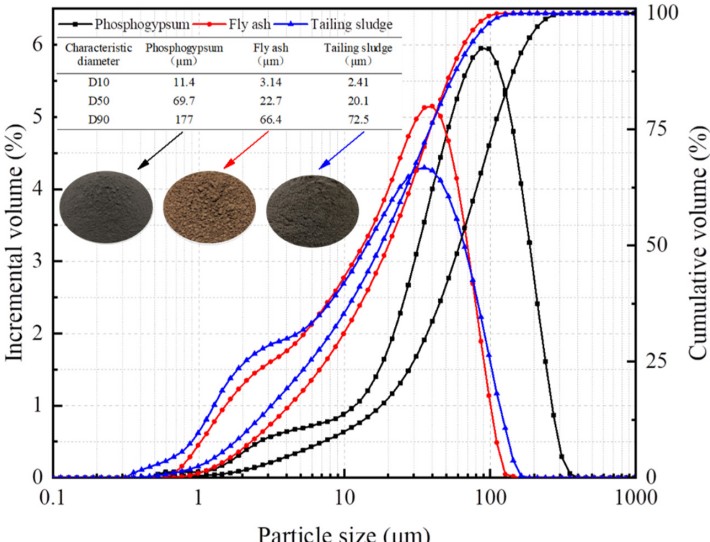

**Figure 1.** Particle size distribution of phosphogypsum, fly ash, and tailing sludge.

**Table 1.** Particle size distribution of tailings.

| Particle Size/mm | Mass Fraction/% | Mass Accumulation/% |
|---|---|---|
| 0.28–1.25 | 26 | 26 |
| 1.25–2 | 27 | 53 |
| 2–3 | 20 | 73 |
| 3–4 | 17 | 90 |
| 4–5 | 10 | 100 |

**Table 2.** Physical properties of test materials.

| Sample | Apparent Density/(kg·m$^{-3}$) | Packing Density/(kg·m$^{-3}$) | Surface Moisture Content/% |
|---|---|---|---|
| Tailings | 2626 | 1464 | 0.120 |
| Tailing sludge | 2653 | 923 | 0.974 |
| Fly ash | 1990 | 650 | 0.049 |
| Phosphogypsum | 1992 | 850 | 8.11 |

**Table 3.** Mineral composition of test materials.

| Sample | Mass Fraction/% | | | | | | | | | | |
|---|---|---|---|---|---|---|---|---|---|---|---|
| | Hydroxyllapatite | Quartz | Hematite | Albite | Plagioclase | Muscovite | Illite | Dolomite | Plaster | Amphibole | Calcite |
| Tailings | 10.15 | 6.91 | 12.75 | - | - | - | - | 69.65 | 0.2 | - | 0.34 |
| Fly ash | - | 61.55 | 1.46 | 15.99 | - | 20.99 | - | - | - | - | - |
| Phosphogypsum | - | 1.35 | - | - | - | - | 4.05 | - | 94.02 | 0.59 | - |
| Tailing sludge | 60.94 | 2.24 | - | 8.38 | 11.42 | - | 9.76 | 6.30 | 0.34 | - | 0.61 |

**Table 4.** Mix weight proportion.

| Group | Phosphogypsum Content/% | Lime Content/% | Mass Percentage | Cement Tailings Ratio |
|---|---|---|---|---|
| B | 0 | 0 | | |
| A1 | 20 | 0 | | |
| A2 | 20 | 0.2 | | |
| A3 | 20 | 1 | | |
| A4 | 20 | 1.8 | 80% | 1:6 |
| A5 | 20 | 2.6 | | |
| A6 | 20 | 3.4 | | |
| A7 | 20 | 4.2 | | |

Note: The percentages are all mass percentages.

2.2.2. Testing Procedure

NMR test: a MesoMR23-060H NMR instrument made by Suzhou Niumag Analytical Instrument Corporation, (Suzhou, China) was used. The samples to be tested were first saturated with water, then wrapped with fresh-keeping film, and last tested by NMR.

UCS test: a No. 01000405 pressure testing machine made by Changlu, Ji'nan Zhongluchang Testing Machine Co., Ltd., (Jinan, China), was used for the UCS test. The sample was placed in the center of the pressure plate and the loading rate was set at 0.2 kN/s while the UCS test was conducted. The UCS value of the sample was calculated. The reference standard for this test is ASTM D2166/D2166M-16.

SEM test: a JSM-IT500LV SEM manufactured by JEOL Ltd. (Japan Electronics, Tokyo, Japan) was used. Samples were taken at the center of the damaged backfill sample, then dried and sprayed with gold, and then observed by SEM to obtain images at different magnifications.

The samples were prepared according to the experimental design, then cured in the curing box, and final tested at the 28-day age. The experimental process is shown in Figure 2.

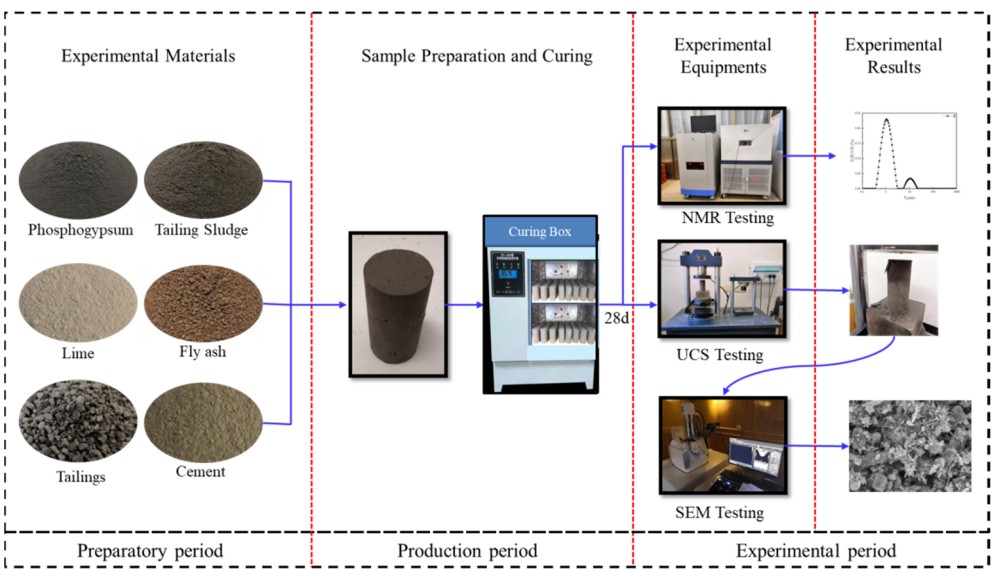

**Figure 2.** Experimental flowchart.

## 3. Results

*3.1. Strength Attributes*

Strength is an essential mechanical characteristic of the backfill and the main control index of the backfill effect. The UCS of lime modified phosphogypsum cemented backfill (LMPGCB) was tested by the uniaxial compression tests, and the results are shown in Figure 3.

It can be seen from Figure 3 that the overall strength first increases and then decreases. The strength decreases exponentially with the increase of lime content from group A2 to group A7, indicating that the addition of lime has a significant influence on the strength. A comparison between group B and group A1 showed that the strength of group A1 increased slightly, indicating that the addition of a certain amount of phosphogypsum is beneficial to the strength. Comparing group A1 with group A2, it can be found that the strength of group A2 increased, which may be attributed to the beneficial effects of lime. Lime and phosphogypsum participate in the hydration reaction, resulting in the greater formation of hydration products [18]. The hydration products have a certain filling effect on the pores and can bond other substances together, thus increasing the strength. When lime is added in high amounts, the volume expands because of the consumption of lime, resulting in increased pore volume, so the strength declines [19].

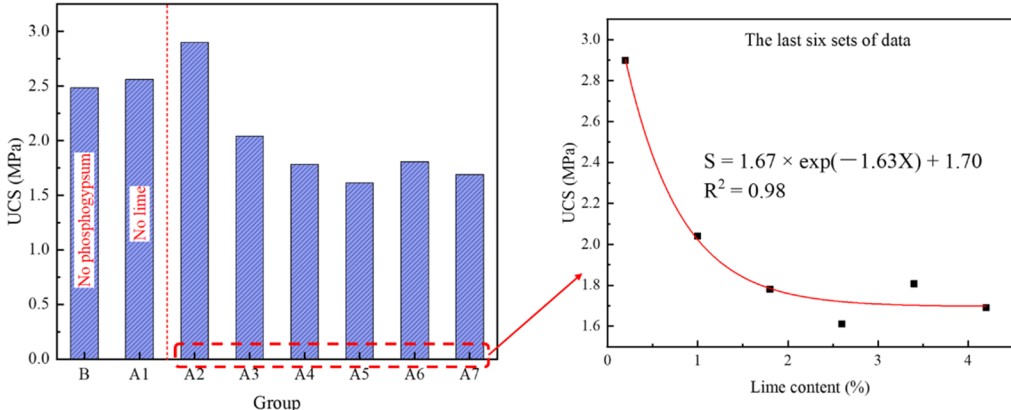

**Figure 3.** Strength characteristics of different groups.

### 3.2. NMR Characteristics

NMR is a nondestructive testing method that can be used to conduct pore testing while maintaining sample integrity. The test results are shown in Figure 4. As there is a certain relationship between pore size and the $T_2$ value, the pore size distribution can be expressed in terms of $T_2$. Their relationship can be expressed as Equation (1) [20,21]:

$$r \approx T_2\rho_2F_s = CT_2 \tag{1}$$

where $r$ represents pore radius (μm), $T_2$ represents the transverse relaxation time (ms), $C$, $\rho_2$, and $F_s$ are constants, $\rho_2$ is the transverse surface relaxation strength (μm/ms), and $Fs$ represents the pore shape.

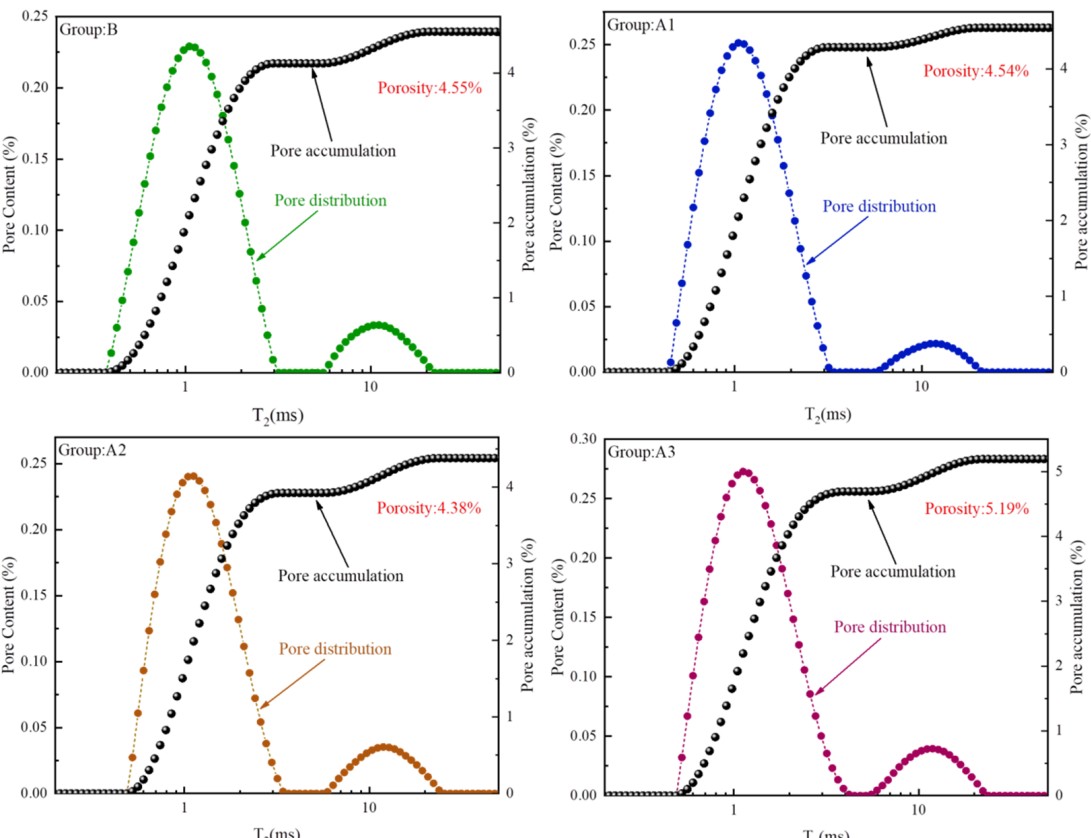

**Figure 4.** *Cont.*

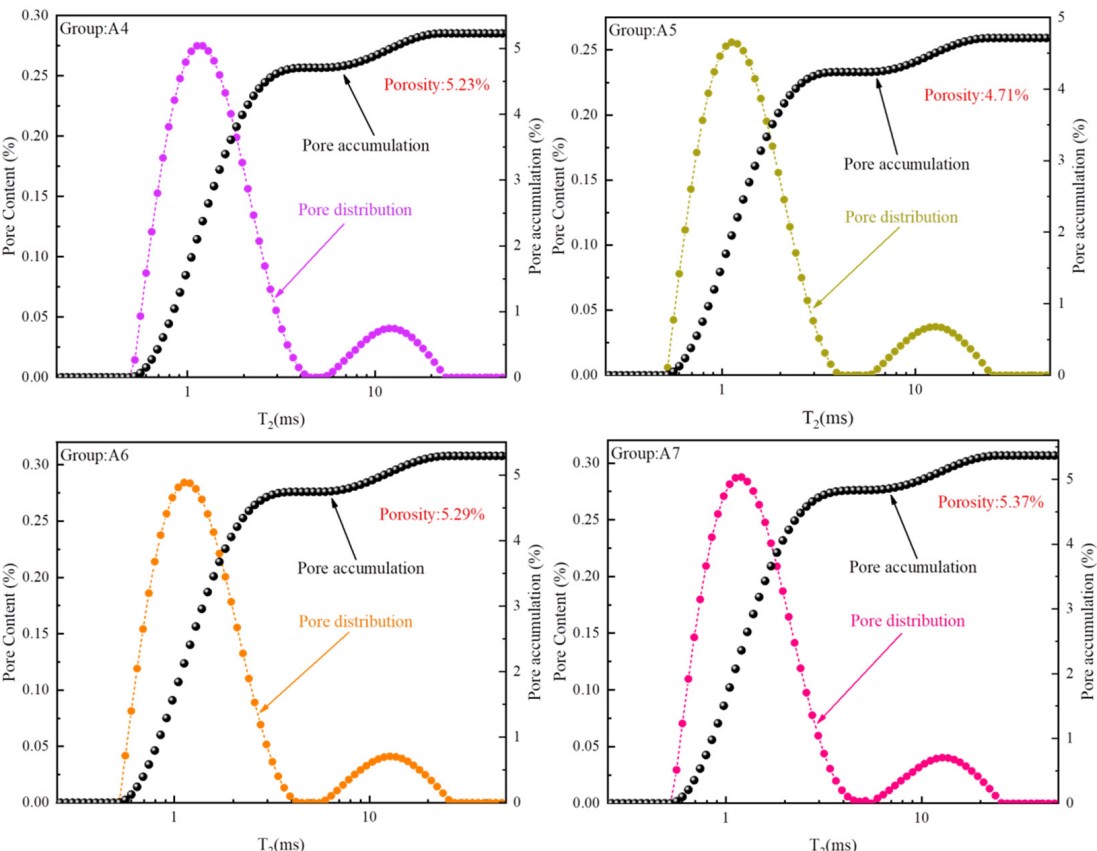

**Figure 4.** Pore distribution of each group LMPGCB.

According to Equation (1), it can be concluded that there is a direct linear relationship between pore size and $T_2$. In Figure 4, the pores are mainly divided into two types. The peak on the left represents the distribution of small pores, and the peak on the right represents the distribution of large pores. In addition, according to the proportion of each peak area of the $T_2$ spectrum that represents the proportion of pores [22], it can be seen that the volume of the small pore is the largest.

As can be seen from Figure 4, porosity as a whole presents an increasing trend in all specimens, but the pore content changes. Comparing group B and group A1, it can be ascertained that there is little difference in their porosity. However, the pore content of the most probable pore size of the small pore of group B is less than 0.25%, whereas the pore content of the most probable pore size of the small pore of group A1 is greater than 0.25%. The pore content of each pore diameter of the large pores of group B is higher than that of group A1. This may be attributed to the phosphogypsum added to group A1, which has an excellent pore filling effect. Comparing group A1 and group A2, it can be found that the porosity of group A2 decreases. After adding lime in group A2, part of phosphogypsum and lime participate in the hydration reaction, and the hydration products have a filling effect on the pores. From group A2 to A7, the porosity tends to increase, as too much lime will expand the volume in consumption, affecting the pores and increasing the porosity.

### 3.3. SEM Characteristics

SEM is a technique to observe the microstructure at high magnification levels. It can observe the microscopic composition of materials at different magnifications. If it is binarized for qualitative analysis, the distribution of pores can be observed. The results are shown in Figure 5.

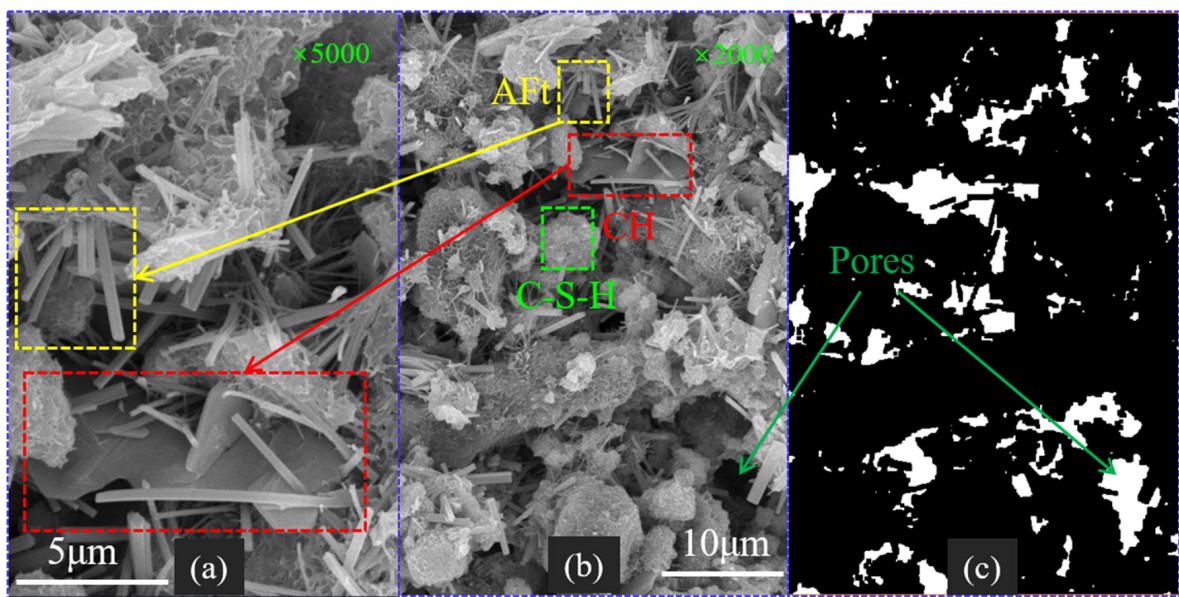

**Figure 5.** SEM image of LMPGCB of group B; (**a**) ×5000 magnification, (**b**) ×2000 magnification, and (**c**) binarization image.

The hydration product can be identified according to the morphology of hydration products [23]. Ettringite (AFt) is mainly in the form of rod-like and needle-like crystals. The gels (C-S-H) are mainly reticular and gelatinous. Calcium hydroxide (CH) is mainly a hexagonal plate shape. Figure 5b shows the products are mainly AFt and C-S-H, with a small amount of CH. Figure 5c is the binarization figure of Figure 5b. The principle of binarization is to transform the solid phase in the SEM image into a black area and the pores into a white area through the gray threshold level [24]. It can be seen from Figure 5 that the pore distribution is relatively concentrated, and the pore shapes are different.

Noise reduction treatment and threshold segmentation are needed before SEM images binarization treatment. First, the median filter is used to reduce the noise of the original SEM image (Figure 6a), and the result is shown in Figure 6b, and morphological opening is used to smooth the median filter noise reduction image, and the result is shown in Figure 6c. Then, the brightness interactive threshold segmentation is carried out, and the result is shown in Figure 6d. Finally, the binarization treatment is carried out, and the result is shown in Figure 6e. The treatment process of SEM is shown in Figure 6.

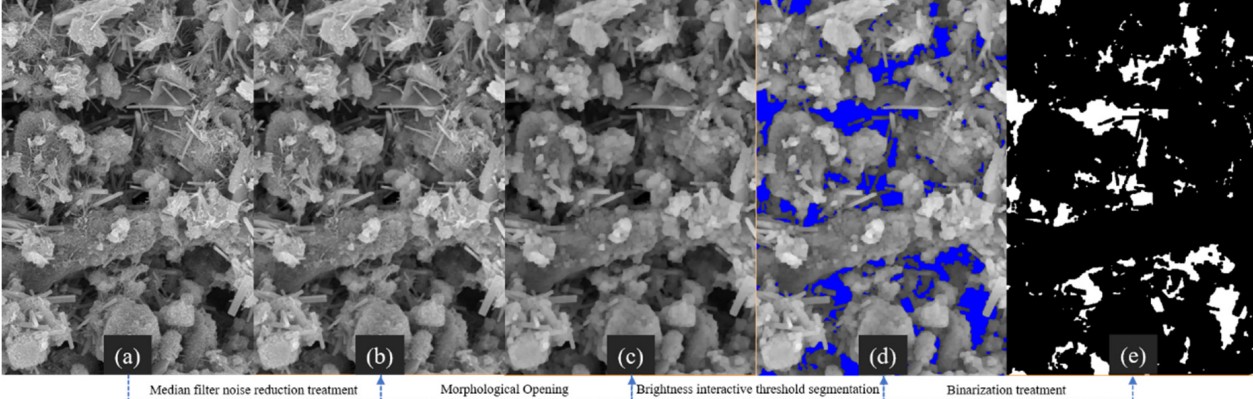

**Figure 6.** The treatment process of SEM image; (**a**) the original SEM image, (**b**) median filter noise reduction image, (**c**) morphological opening image, (**d**) the brightness interactive threshold segmentation image, and (**e**) binarization image.

## 4. Discussion and Analysis

### 4.1. Fractal Characteristics of NMR

4.1.1. Fractal Function of NMR

The pore shapes in the LMPGCB are different. NMR results indicate that it is mainly divided into two types of pores. If the pores are fractal, fractal theory and NMR are combined to study pores so that the fractal characteristics of each type of pore can be obtained. If the pore size function distribution of the pores in the backfill conforms to $N = \propto r^{-D}$, the number of pores with pore size greater than $r$ in the backfill according to the correlation method of calculation dimension can be expressed as Equation (2) [13,17,25]:

$$N = \int_{r}^{r_{max}} P(r)dr = a \times r^{-D} \tag{2}$$

where $r_{max}$ = maximum pore size, $P(r)$ = pore size distribution function, $a$ = proportional constant, and $D$ = fractal dimension.

Taking the derivative of both sides of Equation (2), the expression for $P(r)$ can be obtained as Equation (3):

$$P(r) = \frac{dN}{dr} = -Da \times r^{-D-1} \tag{3}$$

The cumulative volume of pores with a pore size less than $r$ is expressed as Equation (4):

$$V = \int_{r_{min}}^{r} P(r)Ar^3 dr \tag{4}$$

where $A$ is the constant related to pore shape (when the pore is square, $A = 1$, when the pore is circular, $A = 4\pi/3$), and $r_{min}$ is the minimum pore size. Substituting Equation (3) into Equation (4), Equation (5) is obtained:

$$V = -\frac{DaA}{3-D}\left(r^{3-D} - r_{min}^{3-D}\right) \tag{5}$$

When $r$ is the maximum, the total pore volume can be obtained as:

$$Vs = -\frac{DaA}{3-D}\left(r_{max}^{3-D} - r_{min}^{3-D}\right) \tag{6}$$

By combining Equations (5) and (6), the cumulative pore volume fraction with a pore size less than $r$ can be obtained as Equation (7):

$$S_v = \frac{V}{Vs} = \frac{r^{3-D} - r_{min}^{3-D}}{r_{max}^{3-D} - r_{min}^{3-D}} \tag{7}$$

Since $r_{max}$ is much greater than $r_{min}$, Equation (7) can be simplified to Equation (8):

$$S_v = \frac{r^{3-D}}{r_{max}^{3-D}} \tag{8}$$

The pore size expression is also involved in NMR, that is:

$$\frac{1}{T_2} = F_s \frac{\rho_2}{r} \tag{9}$$

Substituting Equation (9) into Equation (8), the Equation (10) is obtained as:

$$S_v = \frac{T_2^{3-D}}{T_{2max}^{3-D}} \tag{10}$$

Taking logarithms on both sides of Equation (10):

$$\lg(S_v) = (3 - D)\lg(T_2) + (D - 3)\lg(T_{2max}) \tag{11}$$

where $S_v$ is the percentage between cumulative pore volume with a transverse relaxation time less than $T_2$ and total pore volume (%), and $T_{2max}$ is the $T_2$ value corresponding to the maximum pore radius (ms).

It can be found from Equation (11) that if lg ($S_v$) and lg ($T_2$) conform to the linear relationship, the pores have fractal properties. The fractal dimension can be calculated according to Equation (12) (the least square method) when the pores are fractal:

$$D = 3 - \frac{\sum_{i=1}^{n} x_i y_i - n\overline{xy}}{\sum_{i=1}^{n} x_j^2 - n\overline{x}^2} = 3 - k \tag{12}$$

4.1.2. NMR-FT Characteristic Analysis of LMPGCB

Taking group B as an example, the relationship between fractal characteristics and the $T_2$ spectrum is established, as shown in Figure 7. It can be seen from Figure 7 that the goodness of fit of $L_1$ and $L_2$ are greater than 0.7, indicating that they have better pore fractal characteristics for small and large pores. However, the goodness of fit of $L_3$ is only 0.53, indicating that the fractal characterization of overall pores using NMR-FT is poor. Furthermore, the fractal dimension of the large pores is larger than that of the small pore, indicating that the large pore is more complex than the small pore.

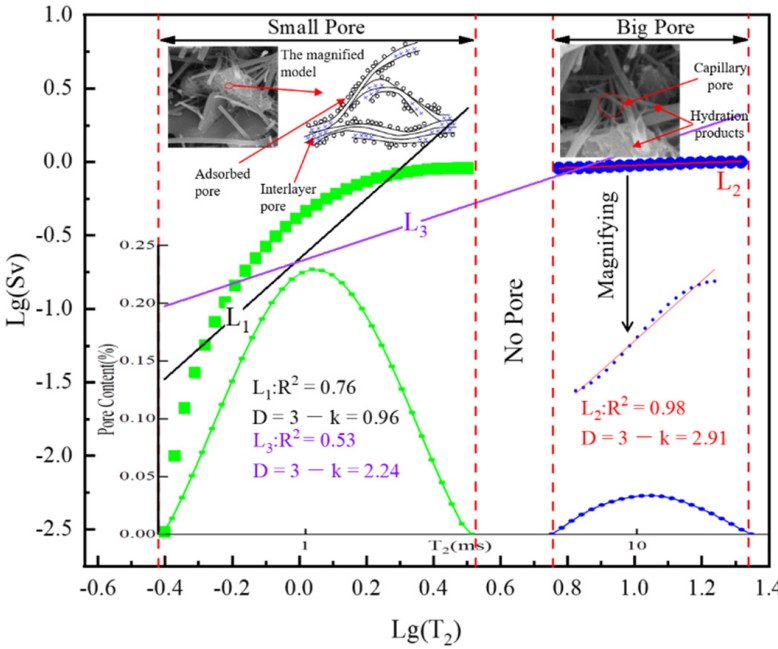

**Figure 7.** Relationship between fractal characteristics and $T_2$ spectrum.

Because of the above results, the fractal theory was used to analyze the small and large pores of LMPGCB, and the complexity of pores was obtained. Figures 7 and 8 show the fractal characteristics of representative groups and fractal dimension distribution of pores. It can be seen from Figure 8 that the fractal characteristics of large and small pores are generally good ($R^2 \geq 0.98$ for large pores and $R^2 \geq 0.7$ for small pores).

Comparing group B and group A1, it can be ascertained that the fractal dimensions of small pores decrease while that of large pores increases. This phenomenon indicates that the complexity of small pores in group A1 decreases while that of large pores increases. This may be attributed to the phosphogypsum addition in group A1, which reduces the cement amount. However, phosphogypsum also has a retarding effect [26], resulting in

reduced hydration products, thereby making the filling effect of small pores better and the filling effect of large pores worse. Thus, the complexity of small pores is weakened, whereas the complexity of large pores is enhanced.

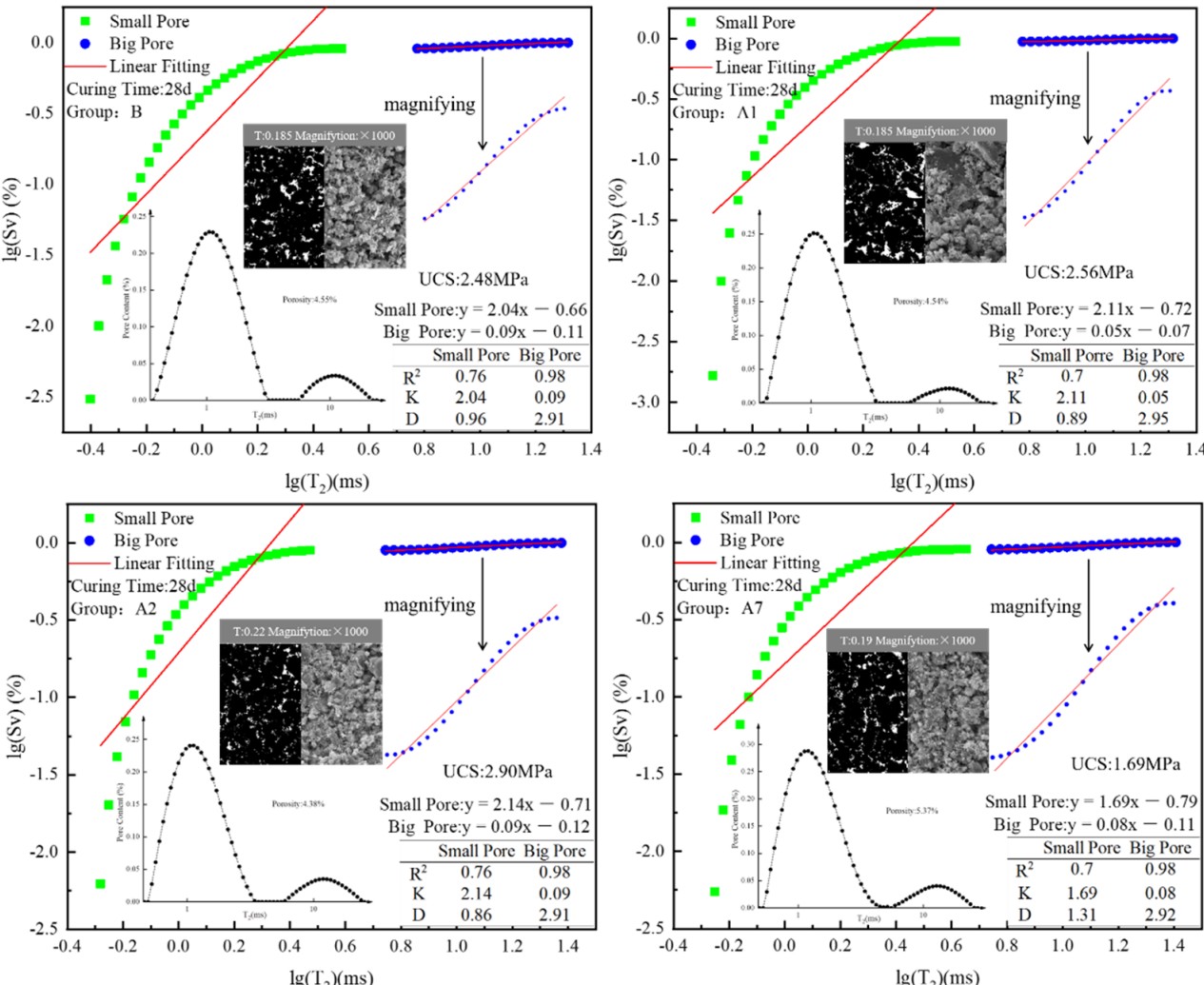

**Figure 8.** Fractal characteristics of representative groups.

Comparing group A1 and group A2, it can be found that the fractal dimensions of both small pores and large pores are reduced, indicating that the complexity of both small pores and large pores in group A2 is weakened. This is because adding lime in group A2 activates the potential activity of phosphogypsum and makes it participate in the hydration reaction, resulting in increased hydration products and a better pore filling effect to reduce the complexity.

From the comparison of group A2 and group A7, it can be found that the fractal dimension of all pores increases, though the increase for large pores is minor. It points out that the complexity of small pores increases while that of big pores increases slightly in group A7. Too much lime enhances hydration reaction, resulting in poor pore filling and transforming large pores into smaller ones. As a result, the complexity of small pores increases significantly, whereas that of large pores changes slightly.

As shown in Figure 9, the fractal dimension of the small pore changes greatly (the range R and standard deviation σ of the fractal dimension of small pores are larger than R and σ of big pores), whereas the overall fractal dimension decreases first and then increases.

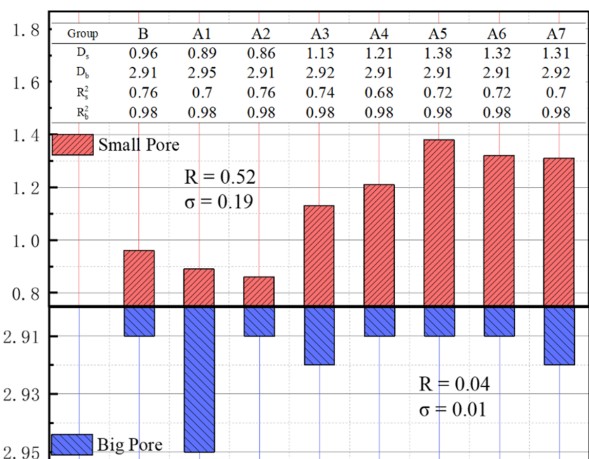

| Group | B | A1 | A2 | A3 | A4 | A5 | A6 | A7 |
|---|---|---|---|---|---|---|---|---|
| $D_s$ | 0.96 | 0.89 | 0.86 | 1.13 | 1.21 | 1.38 | 1.32 | 1.31 |
| $D_b$ | 2.91 | 2.95 | 2.91 | 2.92 | 2.91 | 2.91 | 2.91 | 2.92 |
| $R_s^2$ | 0.76 | 0.7 | 0.76 | 0.74 | 0.68 | 0.72 | 0.72 | 0.7 |
| $R_b^2$ | 0.98 | 0.98 | 0.98 | 0.98 | 0.98 | 0.98 | 0.98 | 0.98 |

**Figure 9.** Fractal dimension distribution of pores.

### 4.2. Fractal Characteristics of SEM

#### 4.2.1. Fractal Principle of SEM

Box dimension is a method of dividing plane structure by square grid, which can quantitatively analyze the complexity of plane structure. The binarization image of SEM is analyzed by box dimension, thus the complexity of pores can be calculated. Box dimension is a criterion for evaluating pore complexity. The larger the box dimension of pores, the higher the complexity of pores [23]. Figure 10 shows the treatment process of SEM images by box dimension. Figure 10a is the flow chart and Figure 10b is the treatment process diagram. The box dimension is that covering the graph by using many small squares. When the side length of small squares is small enough, there is a certain relationship between them, expressed as Equation (13) [27–29]:

$$D = \lim_{r \to 0} \frac{\log\left(N_{(r)}\right)}{-\log(r)} \tag{13}$$

where $D$ represents the box dimension, $r$ represents the box side length, and $N_{(r)}$ represents the number of boxes (a function on box side length $r$).

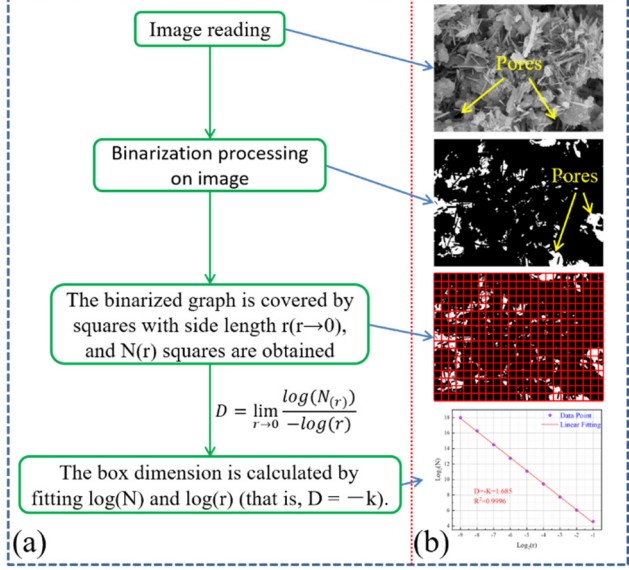

**Figure 10.** The treatment process of SEM image by box dimension. (**a**): the flow chart (**b**): the treatment process diagram.

It can be seen from Equation (13) that when $\log(N_{(r)})$ and $\log(r)$ are linearly fitted, the negative of the slope is the box dimension $D$. As can be seen from Figure 10b, after processing the SEM image (group B ×3000 as an example), the final linear fitting effect of $\log(N_{(r)})$ and $\log(r)$ is good ($R^2 = 0.9996$).

### 4.2.2. SEM-FT Characteristic Analysis of LMPGCB

There are many influencing factors when using box dimensions for image analysis. While studying the influence of various factors on the box dimensions, some researchers found that the magnification had a great influence on it, and it was not appropriate for the magnification to be too large or too small [27,30]. Based on the previous research findings, the magnification levels selected in this study were ×500, ×1000, and ×2000.

Figure 11 is the fitting diagram of $\log(N_{(r)})$ and $\log(r)$ of group B at three magnification levels, while Figure 12 is the box dimension of each group for three magnifications. Figure 11 shows good fitting ($R^2 \geq 0.9996$) at three magnifications, and the box dimension decreases with the increase of the magnification.

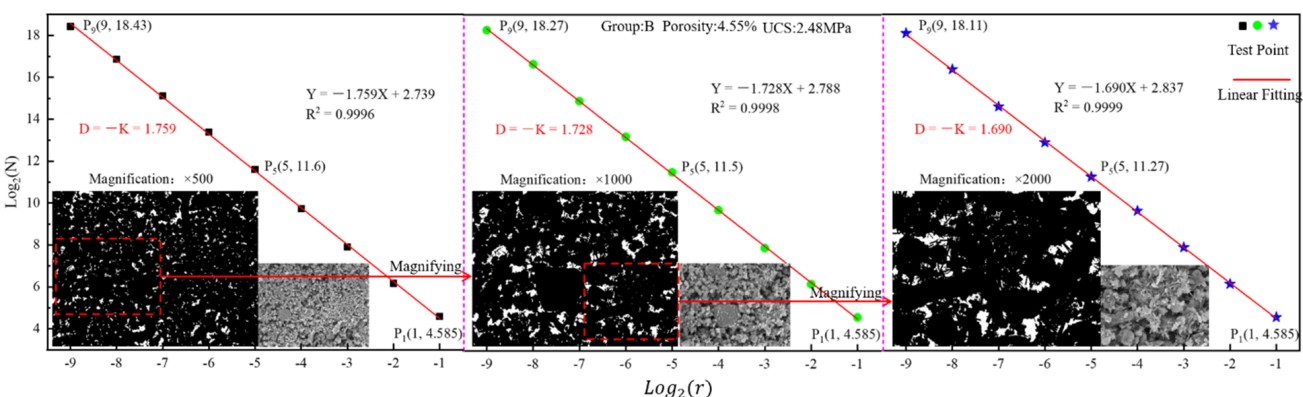

**Figure 11.** The fitting diagram of $\log(N_{(r)})$ and $\log(r)$ of group B at three magnification levels.

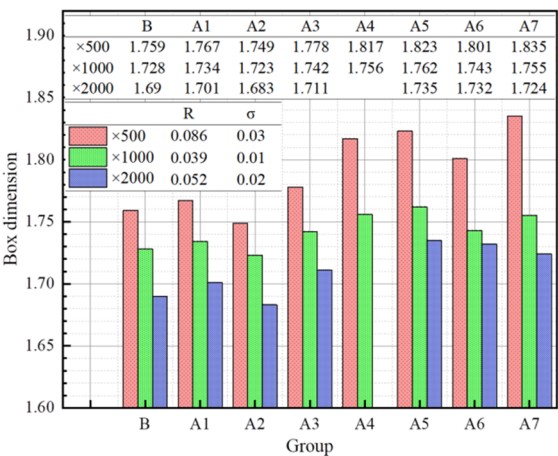

| | B | A1 | A2 | A3 | A4 | A5 | A6 | A7 |
|---|---|---|---|---|---|---|---|---|
| ×500 | 1.759 | 1.767 | 1.749 | 1.778 | 1.817 | 1.823 | 1.801 | 1.835 |
| ×1000 | 1.728 | 1.734 | 1.723 | 1.742 | 1.756 | 1.762 | 1.743 | 1.755 |
| ×2000 | 1.69 | 1.701 | 1.683 | 1.711 | | 1.735 | 1.732 | 1.724 |

| | R | σ |
|---|---|---|
| ×500 | 0.086 | 0.03 |
| ×1000 | 0.039 | 0.01 |
| ×2000 | 0.052 | 0.02 |

**Figure 12.** The box dimension of each group at three magnification levels.

It can be seen from Figure 12 that the larger the magnification, the smaller the box dimension. Shi et al. [30] also found that the box dimension decreased with the increase of magnification when studying the influencing factors of the box dimension of SEM. This is because the object features are magnified at higher magnification; also, the complexity is reduced when the structural features are less comprehensive. Thus, the box dimension is reduced. Figure 12 also indicates that the box dimension decreases first and then increases with the change of group, owing to the disparity in the degree of hydration, which results in different pore filling effects. Thus, the pores are changed.

When magnification is 500 times, its range R and standard deviation σ are both maximum (R: 0.086 > 0.052 > 0.039; σ: 0.03 > 0.02 > 0.01). It also confirms that when the magnification is small, the structural characteristics are more comprehensive, and the difference of samples can be better reflected.

### 4.3. Dimensions Relationship

NMR-FT fractal dimension obtained by combining fractal theory with NMR is only applicable to different types of pores. The SEM-FT box dimension obtained by combining fractal theory with SEM is only applicable to the whole. The two dimensions are fitted to find the functional relationship between them. Figure 13 shows the corresponding functional relationship. Due to NMR-FT, the fractal dimension of large pores changes slightly. Therefore, only the NMR-FT fractal dimension of small pores is used. The specific functional relationship is as follows:

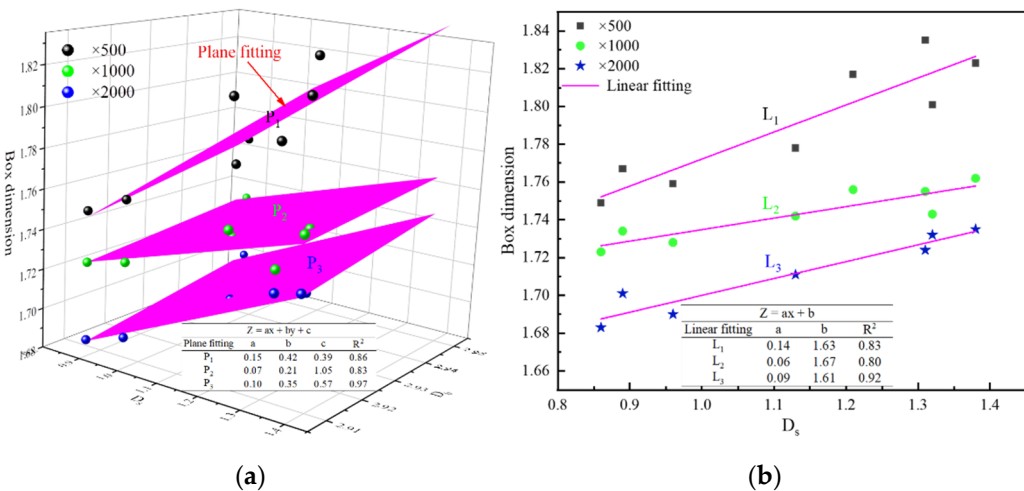

|   | (a) | | (b) |

**Figure 13.** Relationship between NMR-FT fractal dimension and SEM-FT box dimension. (**a**): Relationship between SEM-FT box dimension and each type of pore NMR-FT fractal dimension. (**b**): Relationship between SEM-FT box dimension and small pore NMR-FT fractal dimension.

For each type of pore, the dimension relationship is as follows:
×500 magnification:

$$Z = 0.15x + 0.42y + 0.39 \tag{14}$$

×1000 magnification:

$$Z = 0.07x + 0.21y + 1.05 \tag{15}$$

×2000 magnification:

$$Z = 0.10x + 0.35y + 0.57 \tag{16}$$

where Z represents the SEM-FT box dimension, x represents the NMR-FT fractal dimension of the small pore, and y represents the NMR-FT fractal dimension of the large pore.

For small pore, the dimension relationship is as follows:
×500 magnification:

$$Z = 0.14x + 1.63 \tag{17}$$

×1000 magnification:

$$Z = 0.06x + 1.67 \tag{18}$$

×2000 magnification:

$$Z = 0.09x + 1.61 \tag{19}$$

As can be seen from Figure 13, there is a planar relationship between the NMR-FT fractal dimension of various types of pores and the SEM-FT box dimension. There is a linear relationship between the NMR-FT fractal dimension of small pores and the SEM-FT

box dimension. By comparing the above two types of function relations, it can be seen that the coefficient in front of small pores is similar. It indicates that the dimension of large pores has little influence on them, mainly reflected in a functional relationship between the dimension of small pores and the box dimension. This is because the content of small pores in the backfill material is the largest and plays a significant role.

### 4.4. Relationship Model Construction of Macro-Meso Parameters

#### 4.4.1. The Relationship between Strength and Pore Content

Pores are an important factor affecting the strength properties and, thus, pore content reflects the strength to a certain extent. Figure 14 shows the relationship between strength and pore content. It can be seen that there is a linear inverse relationship between strength and pore content. The relationship between strength and small pore content and porosity is strong ($R^2 \geq 0.95$), whereas the relationship between strength and large pore content is weak ($R^2 = 0.47$). The functional relationship between them is given in Equations (20) and (21).

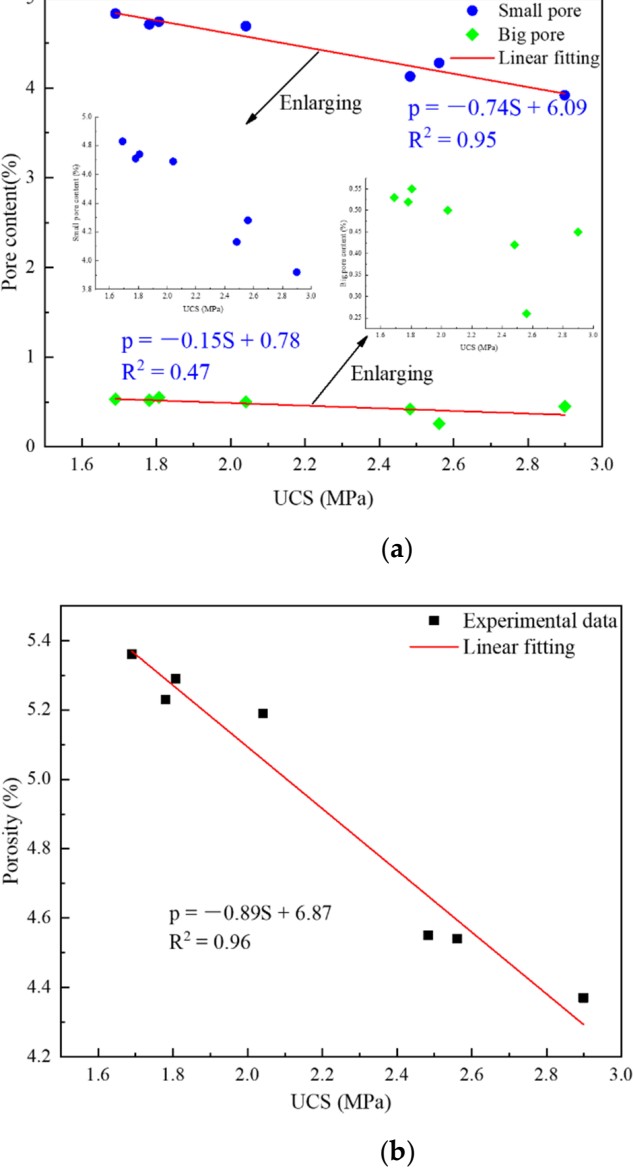

**Figure 14.** The relationship between UCS and pore content. (**a**): Relationship between UCS and small pore content (**b**): Relationship between UCS and porosity.

For small pore:

$$p = -0.74S + 6.09 \tag{20}$$

For porosity:

$$p = -0.89S + 6.87 \tag{21}$$

where p represents pore content (%) and S represents strength (MPa).

Figure 14 shows that the strength decreases linearly with the pore content, indicating that the greater the number of pores, the smaller the strength. More pore content means a lower degree of compactness and reduced density of the backfill material, resulting in reduced mechanical properties. The relationship between large pore content and strength is weak because the large pore content is less, which plays a secondary role in strength development. Therefore, the impact on the strength is weakened.

### 4.4.2. The Relationship between Dimension and Pore Content

The pore content reflects the complexity of pores to a certain extent, and the dimension also reflects the complexity of pores. Therefore, the relationship between pore content and dimension is established. Figure 15 shows the functional relationship between pore content and dimension. As can be seen from the figure, there is a linear directly proportional relationship between them, as follows:

$$x = 0.51p - 1.17 \tag{22}$$

×500 magnification:

$$Z = 0.07p + 1.46 \tag{23}$$

×1000 magnification:

$$Z = 0.03p + 1.61 \tag{24}$$

×2000 magnification:

$$Z = 0.04p + 1.51 \tag{25}$$

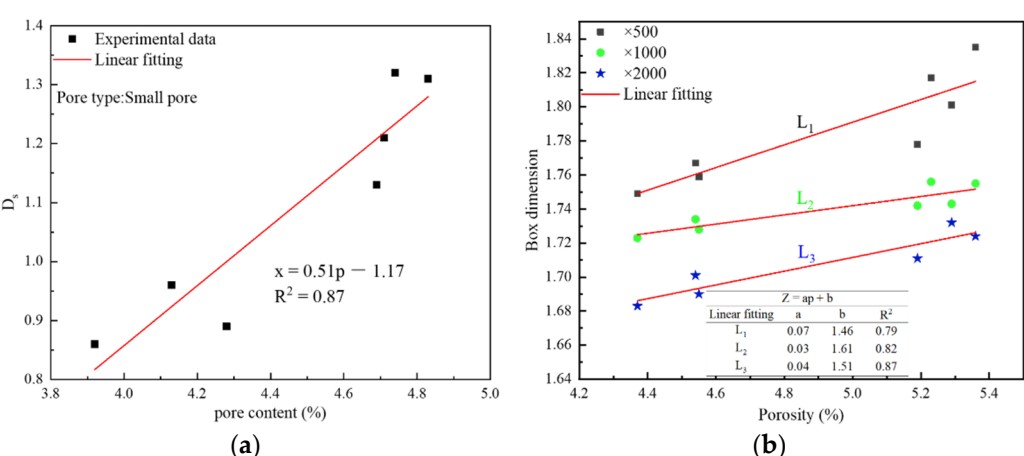

**Figure 15.** The functional relationship between pore content and dimension. (**a**): Relationship between pore content and small pore NMR-FT fractal dimension (**b**): Relationship between porosity and SEM-FT box dimension.

It can be seen from Figure 15 that their fitting effect is good ($R^2 \geq 0.79$), indicating that there is a particular functional relationship between them. Furthermore, the more pore content, the larger the dimension. It can also be seen from Figure 15a that the increase of small pore content leads to the increase of dimension, indicating that the amount of pore content is also an index to evaluate pore complexity. This is because the more pores and more pore shapes indicate the greater complexity. As shown in Figure 15b, the box dimension at the three magnification levels increases with the increase in porosity, but the increase is relatively slow. This is because the increase of porosity

leads to the interconnectivity of large pores, which weakens the influence of small pore content on the dimension so that the overall dimension change is small. Furthermore, the higher the magnification, the better the fitting effect, which is because better pixels and more prominent microstructure are observed at higher magnification. However, the magnification should not be too large. If the magnification is too large, the characterization of the structural characteristics of the sample is not comprehensive [24].

### 4.4.3. The Relationship between Dimension and Strength

The dimension reflects the complexity of pores, and the complexity of pores affects the strength. Thus, the dimension also reflects the strength attributes to a certain extent. Figure 16 shows the functional relationship between strength and dimension. As shown in Figure 16, there is a linear inverse relationship between them, as given in Equation (26) through Equation (29):

$$S = -2.25x + 4.66 \tag{26}$$

$\times 500$ magnification:

$$S = -13.89Z + 27 \tag{27}$$

$\times 1000$ magnification:

$$S = -31.97Z + 57.83 \tag{28}$$

$\times 2000$ magnification:

$$S = -23.16Z + 41.79 \tag{29}$$

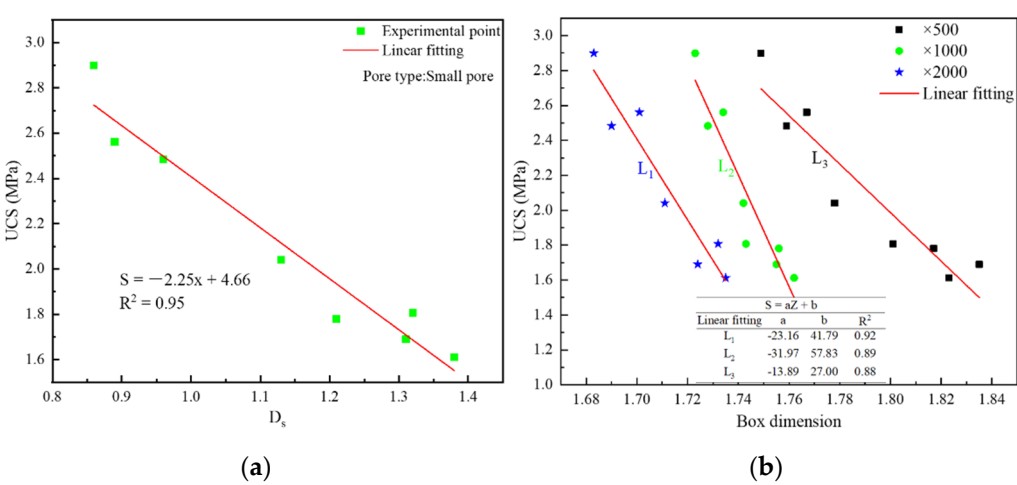

|  | (a) | (b) |

**Figure 16.** The functional relationship between strength and dimension. (**a**): Relationship between UCS and small pore NMR-FT fractal dimension(**b**): Relationship between UCS and SEM-FT box dimension.

The strength decreases with the increase of dimension, and the fitting effect is good ($R^2 \geq 0.88$). It shows that the change of dimension can reflect the change of strength under certain conditions. This is because the change of dimension indicates that changes in pore structure, thus affecting the strength. With the increase of dimension, the pore complexity increases while the filling effect of hydration products and degree of compactness becomes poor, resulting in a corresponding strength reduction. As shown from Figure 16a, the change of the dimension of the small pore can well reflect the strength change. This is because the small pores account for most of the total pores; thus, the change of small pores significantly influences the overall change. At three levels of magnification, the strength decreases with the dimension increases (Figure 16b).

### 5. Conclusions

In this paper, based on the mechanical tests on lime modified phosphogypsum cemented backfill (LMPGCB), the microstructure of the LMPGCB sample was characterized

by nuclear magnetic resonance (NMR) and scanning electron microscopy (SEM) imaging. Then the fractal theory was used for in-depth analysis. The fractal characteristics of various types of pores in the backfill material, the overall fractal characteristics, and their relationship were obtained. The functional relationship model of strength was established by using fractal characteristics. The main conclusions drawn from this study are as follows:

(1) The NMR-FT fractal effect of various types of pores is good, but the overall fractal effect is poor. The model of the dimension of the small pores is strong. The fractal dimension of the small pore is between 0.86–1.38 (the overall trend decreases first and then increases).

(2) The SEM-FT box dimension is 1.749–1.835 at ×500 magnification, 1.723–1.762 at ×1000 magnification, and 1.683–1.735 at ×2000 magnification, whose overall trend also decreased first and then increased. With the increase of magnification, the box dimension decreases.

(3) By developing the relationship between NMR-FT fractal dimension and the SEM-FT box dimension, it is seen that there is a direct linear relationship between the SEM-FT box dimension and the NMR-FT fractal dimension of small pores. The relationship between the SEM-FT box dimension and the NMR-FT fractal dimension conforms to plane relationship.

(4) By constructing the functional relationship model between dimension and pore content and strength, it is shown that there is a linearly increasing relationship between dimension and pore content and a linear decreasing relationship between dimension and strength.

**Author Contributions:** Conceptualization, F.Z. and J.H.; validation, F.M.; formal analysis, F.Z. and Y.Y.; investigation, Y.Y.; writing—original draft preparation, F.Z.; writing—review and editing, F.Z., J.H. and H.X.; project administration, J.H.; funding acquisition, J.H. All authors have read and agreed to the published version of the manuscript.

**Funding:** This work was financially supported by the National Key Research and Development Program of China (Grant no. 2017YFC0602901) and the National Natural Science Foundation of China (Grant no. 41672298).

**Data Availability Statement:** The source data can be obtained in the article.

**Acknowledgments:** The authors gratefully acknowledge the support of the National Key Research and Development Program of China through Grant no. 2017YFC0602901 and the support of the National Natural Science Foundation of China (Grant no. 41672298). They are also thankful for the instructional support specialist in Modern Analysis and Testing Central of Central South University.

**Conflicts of Interest:** The authors declare no conflict of interest.

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
