# Peer review of "Cross-Scale Study on Lime Modified Phosphogypsum Cemented Backfill by Fractal Theory"

_minerals, doi:10.3390/min12040403_

Round 1
Reviewer 1 Report
Dear respected author,
The paper presents the interesting experimental research work on lime modified phosphogypsum cemented backfill, but, it contains a few issues that detract from its overall quality as shown below:
- The quality of the English in the paper needs to be improved.
- What is the exact innovation of the work? Please clearly state!
- How did the authors used the sieve analysis for particle size distribution while most particles have the size less than 100 Mm!
- Line 98: “The samples were cured in the curing chamber”. How were the samples cured? Please explain (mention the ASTM).
- The results illustrated on Figure 3 seems to be in contradiction with the results presented on Figure 4! For example, based on Figure 3, group A5 has the lowest UCS, and so it should have the highest porosity! But based on Figure 4, group A2 has the highest porosity! As we know, the less UCS, the higher porosity! The authors should explain about theses contradictions!

Author Response
Dear Editor and Reviewers:
Thank you for your letter and for the reviewers’ comments concerning our manuscript entitled “Cross-scale study on lime modified phosphogypsum cemented backfill by fractal theory”(ID: minerals-1640967). Those comments are all valuable and very helpful for revising and improving our paper, as well as the important guiding significance to our researches. We have studied comments carefully and have made correction which we hope meet with approval. Revised portion are marked in red in the paper. The main corrections in the paper and the responds to the reviewers comments are as flowing:
Responds to the reviewers comments:
(Blue indicates the problem, red indicates the respond.)
1.The quality of the English in the paper needs to be improved.
The English of the whole manuscript has polished thoroughly. The polish certificate has been submitted.
2.What is the exact innovation of the work? Please clearly state!
The innovation of this paper is to explore the influence of fractal characteristics of different pore sizes in the backfill on the overall fractal characteristics through fractal theory, so as to comprehensively analyze the pores in the backfill from local to overall, and try to analyze the influence of fractal on strength.
3.How did the authors used the sieve analysis for particle size distribution while most particles have the size less than 100μm!
The particle size of less than 100 μm is measured by laser particle size analyzer (Mastersizer 2000). This instrument has been added to the paper.
4.Line 98: “The samples were cured in the curing chamber”. How were the samples cured? Please explain (mention the ASTM).
Put the sample into the curing box, and then set the temperature of the curing box at 20℃ and the humidity at 99% for curing.
5.The results illustrated on Figure 3 seems to be in contradiction with the results presented on Figure 4! For example, based on Figure 3, group A5 has the lowest UCS, and so it should have the highest porosity! But based on Figure 4, group A2 has the highest porosity! As we know, the less UCS, the higher porosity! The authors should explain about theses contradictions!
UCS is not only related to porosity, but also related to the proportion of pores with different pore size. According to relevant research, the proportion of big pores is the main factor affecting the UCS. From the pore proportion of group A5, it can be seen that the decrease of porosity is mainly due to the decrease of small pore content, while the content of big pores affecting UCS does not decrease, so the UCS is low.
Special thanks to you for your good comments.

Reviewer 2 Report
Dear author,
I have read you paper with attention and interest.
I general I asses it like text worth of publication however I have got a few comments which I would like to propose to consider.
Abstract - is clear
Introduction - is clear
Fig. 1. and 2. letter elements are to small, and therefore illegible, however most of figures are illegible.
Line 137 references "17-18"
Discussion and analysis - are clear
Conclusions - are clear
Regards
Author Response
Dear Editor and Reviewers:
Thank you for your letter and for the reviewers’ comments concerning our manuscript entitled “Cross-scale study on lime modified phosphogypsum cemented backfill by fractal theory”(ID: minerals-1640967). Those comments are all valuable and very helpful for revising and improving our paper, as well as the important guiding significance to our researches. We have studied comments carefully and have made correction which we hope meet with approval. Revised portion are marked in red in the paper. The main corrections in the paper and the responds to the reviewers comments are as flowing:
Responds to the reviewers comments:
(Blue indicates the problem, red indicates the respond.)
1.Fig. 1. and 2. letter elements are to small, and therefore illegible, however most of figures are illegible.
The figures have been enlarged.
2.Line 137 references "17-18"
This part has been revised.
Special thanks to you for your good comments.

Reviewer 3 Report
The authors are congratulated for the document "Cross-scale study on lime modified phosphogypsum cemented backfill by fractal theory ", it has good structure, as well as in the fulfillment of the objectives, references, and the practical approach on the analysis and methodology of samples based on the fractal theory, however, the only pertinent observations that I will be able to comment are:
Abstract:
Line 22-26 – Describe in a clearer manner the fractal and mathematical results obtained in the analysis.
Introduction:
Expand the information of the fractal theory and its use.
Line 30 – Emphasize the introduction of why the study is generated, give a short and comparative preamble to the use of this technique versus the traditional analysis processes.
Line 53-59 – Review the structure and wording, expand the paragraph information to better understand it.
Line 65-70 – Justify why the use of lime and phosphogypsum are suitable for this type of study, expand the information to better understand it.
Materials and Methods
Line 77-79 – Clarify if the choice of material is not a specific condition for the results.
Figure 1 - Support yourself with a better image and differentiate the internal data for a better reading.
Line 101-112 (2.2.2. Testing procedure)
(Restructure the paragraph – Expand the idea and better identify the arguments, standards, technical parameters, and research guidelines)
Figure 2 – Describe the elements of the experimental process of figure 2.
Results
Line 119-121 (3.1. Strength attributes)
Clarify or expand the paragraph of why the use of uniaxial tests instead of triaxial or load point tests if this was a determinant for the research guidelines.
Line 126-128 – Specify the idea, better clarify the description of the paragraph.
Line 142-159 – Improve the description of the paragraphs, the paragraph is very large since it has a direct reading of the results of figure 4, separate information into lighter paragraphs so that the reading of the results obtained in the investigation is better understood.
Line 178-189 – Improve the description of the paragraphs, the paragraph is very large, separate information into lighter paragraphs to better understand them.
Discussion and analysis
Line 248-271 – Improve the description of the paragraphs, the paragraph is very large, it is recommended to separate it into paragraphs, into lighter elements of information so that the analysis of the research is better understood.
Figure 7 and 8 – Support yourself with a better image and differentiate the internal data for a better reading (the figures are very small)
Line 285-292 – Expand the information in the paragraph to understand it better.
Line 303-322 – Restructure the paragraph.
Line 366-384 – Unify the information and restructure the paragraph
Line 400-415 – Restructure the paragraph, improve its wording.
Conclusion
Line 448-456 – Restructure the paragraph, improve its wording to better understand the obtained conclusions.
References
None
Author Response
Dear Editor and Reviewers:
Thank you for your letter and for the reviewers’ comments concerning our manuscript entitled “Cross-scale study on lime modified phosphogypsum cemented backfill by fractal theory”(ID: minerals-1640967). Those comments are all valuable and very helpful for revising and improving our paper, as well as the important guiding significance to our researches. We have studied comments carefully and have made correction which we hope meet with approval. Revised portion are marked in red in the paper. The main corrections in the paper and the responds to the reviewers comments are as flowing:
Responds to the reviewers comments:
(Blue indicates the problem, red indicates the respond.)
Abstract:
1.Line 22-26 – Describe in a clearer manner the fractal and mathematical results obtained in the analysis.
The abstract part has been revised.
Introduction:
2.Expand the information of the fractal theory and its use.
The relevant content has been supplemented.
3.Line 30 – Emphasize the introduction of why the study is generated, give a short and comparative preamble to the use of this technique versus the traditional analysis processes.
The relevant content has been revised.
4.Line 53-59 – Review the structure and wording, expand the paragraph information to better understand it.
The relevant content has been revised.
5.Line 65-70 – Justify why the use of lime and phosphogypsum are suitable for this type of study, expand the information to better understand it.
This reason has been explained in this revised paper.
Materials and Methods
6.Line 77-79 – Clarify if the choice of material is not a specific condition for the results.
The choice of material is not a specific condition for the results.
7.Figure 1 - Support yourself with a better image and differentiate the internal data for a better reading.
Figure 1 has been enlarged.
Line 101-112 (2.2.2. Testing procedure)
8.(Restructure the paragraph – Expand the idea and better identify the arguments, standards, technical parameters, and research guidelines)
This part has been revised.
9.Figure 2 – Describe the elements of the experimental process of figure 2.
The elements of the experimental process of figure 2 have been described in this revised paper.
Results
Line 119-121 (3.1. Strength attributes)
10.Clarify or expand the paragraph of why the use of uniaxial tests instead of triaxial or load point tests if this was a determinant for the research guidelines.
Because the backfill material is mainly subjected to axial stress in the goaf, this study mainly focuses on the uniaxial compressive strength.
11.Line 126-128 – Specify the idea, better clarify the description of the paragraph.
The paragraph has been revised.
12.Line 142-159 – Improve the description of the paragraphs, the paragraph is very large since it has a direct reading of the results of figure 4, separate information into lighter paragraphs so that the reading of the results obtained in the investigation is better understood.
The paragraph has been revised.
13.Line 178-189 – Improve the description of the paragraphs, the paragraph is very large, separate information into lighter paragraphs to better understand them.
The paragraph has been revised.
Discussion and analysis
14.Line 248-271 – Improve the description of the paragraphs, the paragraph is very large, it is recommended to separate it into paragraphs, into lighter elements of information so that the analysis of the research is better understood.
The paragraph has been revised.
15.Figure 7 and 8 – Support yourself with a better image and differentiate the internal data for a better reading (the figures are very small)
The figure has been enlarged.
16.Line 285-292 – Expand the information in the paragraph to understand it better.
The information in the paragraph has been expanded.
17.Line 303-322 – Restructure the paragraph.
The paragraph has been revised.
18.Line 366-384 – Unify the information and restructure the paragraph
The paragraph has been revised.
19.Line 400-415 – Restructure the paragraph, improve its wording.
The paragraph has been revised.
Conclusion
20.Line 448-456 – Restructure the paragraph, improve its wording to better understand the obtained conclusions.
Some conclusions have been revised.
Special thanks to you for your good comments.

Reviewer 4 Report
The paper covers a topic that is central in the journal scope, as it deals with tests on lime modified phosphogypsum cemented backfill, and the microsctructure of the backfill samples. The microstructure was characterized by nuclear magnetic resonance and by scanning electron microscopy imaging. In addition fractal theory was employed for in-depth analysis. While the conceptual flow drawn by Authors is easy to follow, there are several points that needs to be addressed:
Line 53-54: Please further argue the applicability of fractal theory for pore analysis. Only one reference is used to confirm the applicability of the fractal theory in this study.
Line 67-69: Please explain why these filling materials in particular were used.
Line 69-70: Please indicate the standard/norm according to which the UCS tests were conducted.
Line 79: Please explain why exactly lime with 85% CaO was used?
Line 80: Please indicate the standard/norm according to which the sieve analysis was conducted.
Figure 1 should be enlarged.
Line 95-96: Please further describe the labels B and A. Why B, for example, is not A0?
Table 4. The percentages defined are unclear. Please clarify.
Line 120-121: "...overall strength shows an increasing trend which decreases subsequently." Please reshape the sentence or describe the observed differently.
Line 123-124: "...the strength of group A1 increased." The sentence seems unfinished. Please explain in relation to what the strength of group A1 has increased.
Line 126-127: "Lime and phosphogypsum participate in the hydration reaction, resulting in greater formation hydration products." Please indicate if there is additional confirmation of this in the reviewed literature? Has anyone else perhaps found out about this as well?
Figure 3. I suggest that all graphs be overlapped and displayed in one image or possibly two images. This makes it easier to make comparisons and draw conclusions.
Please check if all the symbols used in expressions are defined after their first appearance.
Figure 15 and Figure 16: The (b) is written in red. Please correct.
In chapter Conclusion under some text there is gray background (it seems as the text is highlighted). This background should be removed.
Line 457-458: The conclusion is unclear. Please rephrase the sentence.
The literature is insufficiently researched. I propose to extend the literature review to the European and North American research area. Additional literature review should be included in Introduction chapter.
Abstract: I suggest removing all equations from the conclusion and simplifying the reviews described through points (1) - (3).
Keywords: I suggest changing your keywords. The first keyword is overly complex and very specific. Also, I suggest omitting abbreviations unless they are generally accepted.
Author Response
Dear Editor and Reviewers:
Thank you for your letter and for the reviewers’ comments concerning our manuscript entitled “Cross-scale study on lime modified phosphogypsum cemented backfill by fractal theory”(ID: minerals-1640967). Those comments are all valuable and very helpful for revising and improving our paper, as well as the important guiding significance to our researches. We have studied comments carefully and have made correction which we hope meet with approval. Revised portion are marked in red in the paper. The main corrections in the paper and the responds to the reviewers comments are as flowing:
Responds to the reviewers comments:
(Blue indicates the problem, red indicates the respond.)
1.Line 53-54: Please further argue the applicability of fractal theory for pore analysis. Only one reference is used to confirm the applicability of the fractal theory in this study.
Relevant references have been added.
2.Line 67-69: Please explain why these filling materials in particular were used.
Because phosphogypsum and lime participate in hydration reaction and can replace part of cement, they are taken as the research object.
3.Line 69-70: Please indicate the standard/norm according to which the UCS tests were conducted.
The reference standard for this test is ASTM D2166/D2166M-16. It has been added to the paper.
4.Line 79: Please explain why exactly lime with 85% CaO was used?
85% CaO is only a physical parameter of lime, not a specific requirement.
5.Line 80: Please indicate the standard/norm according to which the sieve analysis was conducted.
The reference standard for this test is JTG E42-2005. It has been added to the paper.
6.Figure 1 should be enlarged.
Figure 1 has been enlarged.
7.Line 95-96: Please further describe the labels B and A. Why B, for example, is not A0?
In this paper, label B has no special meaning. Group B is a blank control group, so label B is taken.
8.Table 4. The percentages defined are unclear. Please clarify.
The percentages all are mass percentage. It has been added to the paper.
9.Line 120-121: "...overall strength shows an increasing trend which decreases subsequently." Please reshape the sentence or describe the observed differently.
This sentence has been revised in this revised paper.
10.Line 123-124: "...the strength of group A1 increased." The sentence seems unfinished. Please explain in relation to what the strength of group A1 has increased.
The content has been supplemented.
11.Line 126-127: "Lime and phosphogypsum participate in the hydration reaction, resulting in greater formation hydration products." Please indicate if there is additional confirmation of this in the reviewed literature? Has anyone else perhaps found out about this as well?
Relevant reference has been added.
12.Figure 3. I suggest that all graphs be overlapped and displayed in one image or possibly two images. This makes it easier to make comparisons and draw conclusions.
Figure 3 has been revised in this revised paper.
13.Please check if all the symbols used in expressions are defined after their first appearance.
All the symbols used in expressions has been defined uniformly.
14.Figure 15 and Figure 16: The (b) is written in red. Please correct.
They have been revised in this revised paper.
15.In chapter Conclusion under some text there is gray background (it seems as the text is highlighted). This background should be removed.
This background has been removed.
16.Line 457-458: The conclusion is unclear. Please rephrase the sentence.
The sentence has been revised.
17.The literature is insufficiently researched. I propose to extend the literature review to the European and North American research area. Additional literature review should be included in Introduction chapter.
Relevant literature has been added.
18.Abstract: I suggest removing all equations from the conclusion and simplifying the reviews described through points (1) - (3).
In abstract section, I has removed all equations from the conclusion and simplified the reviews.
19.Keywords: I suggest changing your keywords. The first keyword is overly complex and very specific. Also, I suggest omitting abbreviations unless they are generally accepted.
They have been revised in this revised paper.
Special thanks to you for your good comments.

Round 2
Reviewer 4 Report
I have no further comments or suggestions.